# Direct deoxygenative borylation of carboxylic acids

Jianbin Li [1,2,3], Chia-Yu Huang [1,2,3], Mohamad Ataya[1,3], Rustam Z. Khaliullin[1,4✉] & Chao-Jun Li [1,2,4✉]

Carboxylic acids are readily available, structurally diverse and shelf-stable; therefore, converting them to the isoelectronic boronic acids, which play pivotal roles in different settings, would be highly enabling. In contrast to the well-recognised decarboxylative borylation, the chemical space of carboxylic-to-boronic acid transformation via deoxygenation remains underexplored due to the thermodynamic and kinetic inertness of carboxylic C-O bonds. Herein, we report a deoxygenative borylation reaction of free carboxylic acids or their sodium salts to synthesise alkylboronates under metal-free conditions. Promoted by a uniquely Lewis acidic and strongly reducing diboron reagent, bis(catecholato)diboron ($B_2cat_2$), a library of aromatic carboxylic acids are converted to the benzylboronates. By leveraging the same borylative manifold, a facile triboration process with aliphatic carboxylic acids is also realised, diversifying the pool of available 1,1,2-alkyl(trisboronates) that were otherwise difficult to access. Detailed mechanistic studies reveal a stepwise C-O cleavage profile, which could inspire and encourage future endeavours on more appealing reductive functionalisation of oxygenated feedstocks.

[1] Department of Chemistry, McGill University, Montreal, QC, Canada. [2] FRQNT Centre for Green Chemistry and Catalysis, Montreal, QC, Canada. [3]These authors contributed equally: Jianbin Li, Chia-Yu Huang, Mohamad Ataya. [4]These authors jointly supervised this work: Rustam Z. Khaliullin, Chao-Jun Li. ✉email: rustam.khaliullin@mcgill.ca; cj.li@mcgill.ca

Joining carbon and boron, two neighbouring elements on the periodic table, is of fundamental interest due to the widespread utility of the organoboron compounds[1–7]. Among them, alkylboronic acids and their derivatives have gained increasing attention. In the synthetic community, they serve as diversifiable building blocks, constituting numerous modular approaches in cross-couplings, metallate rearrangements and deborylative additions. Their chemical versatility also leads to broad applications spanning functional materials and pharmaceuticals. Especially for the latter, owing to their unique structural traits, boronic functionality could mimic the amide moiety during enzymatic hydrolysis and represent the potent bioisostere of the carboxylic group[8].

Despite these far-reaching applications, access to this class of boron compounds largely relies on chemical synthesis because of their scarcity in nature[1–7]. General methods to forge aliphatic C–B bonds include hydroboration of alkenes and transmetallation of organometallic species. Besides, the breakthrough of catalysed or uncatalysed C–X (X = H[9–13] or (pseudo)halides[14–19]) borylation paradigms significantly reshaped the landscape of boron chemistry, unlocking access to countless previously elusive boron compounds. However, these precedent strategies mostly started from non-renewable or pre-synthesised substrates, which suffered limited availability, perceived instability and sometimes toxicity.

Carboxylic acids are privileged chemical entities in many scientific areas considering their natural abundance, structural diversity and chemical stability. These benign features made them ideal starting materials for the alkylboronate synthesis and encouraged continuing exploration in the chemical space of carboxylic-to-boronic acid transformation (Fig. 1a)[20]. In this context, Baran and his co-workers pioneered the field by developing a Ni-catalysed decarboxylative borylation reaction through repurposing the N-(alkyloxy)phthalimides as redox-active alkyl radical precursors[21]. The same group, collaborating with Blackmond's laboratory, reported two other practical protocols based on the inexpensive copper catalysis and electrochemical means[22,23]. Similarly, Aggarwal and his colleagues disclosed a photoinduced decarboxylative borylation manifold with the same redox-active esters (RAEs), in which the strategic assembly of a light-absorbing boron complex could avoid the usage of transition metal catalysts[24]. Noticeably, all these decarboxylation-driven methods allowed access to complex boronates from some densely functionalised drug molecules, natural products or native peptides.

An equally important alternative to convert carboxylic acids into their boronic analogues is through deoxygenative borylation, ideally in the absence of metal catalysts and exogenous additives. Although multistep sequences could be conceivable to impart the same deoxy-borylation reactivity (Fig. 1b), a straightforward deoxy-borylative transformation of free carboxylic acids using simple and common reagents remained undeveloped, which was in stark contrast to its well-known decarboxylative counterpart. Presumably, this could be attributed to the prohibitively high overall energy barrier for three C–O bond scissions and the frequently problematic carboxylic proton. To this end, a robust and balanced system was required to activate the C–O bonds while controlling the chemoselectivity and circumventing some off-target reactivities (e.g. partial and over reduction). Notwithstanding these substantial challenges, tackling the borylative reduction of carboxylic groups would be highly rewarding, not only enriching the existing borylation toolkits but also providing a library of isoelectronic homo-analogues of carboxylic acids for bioisosteric lead screening and optimisation.

Herein, we would like to document a distinct deoxygenative strategy for synthesising alkylboronates from the ubiquitous carboxylic acids without prior functionalisation and metal catalysts (Fig. 1c). This simple approach could enable a direct deoxygenative borylation of aromatic carboxylic acids, facilitating a streamlined synthesis of benzylboronates with various substitution patterns. The same borylation manifold showed

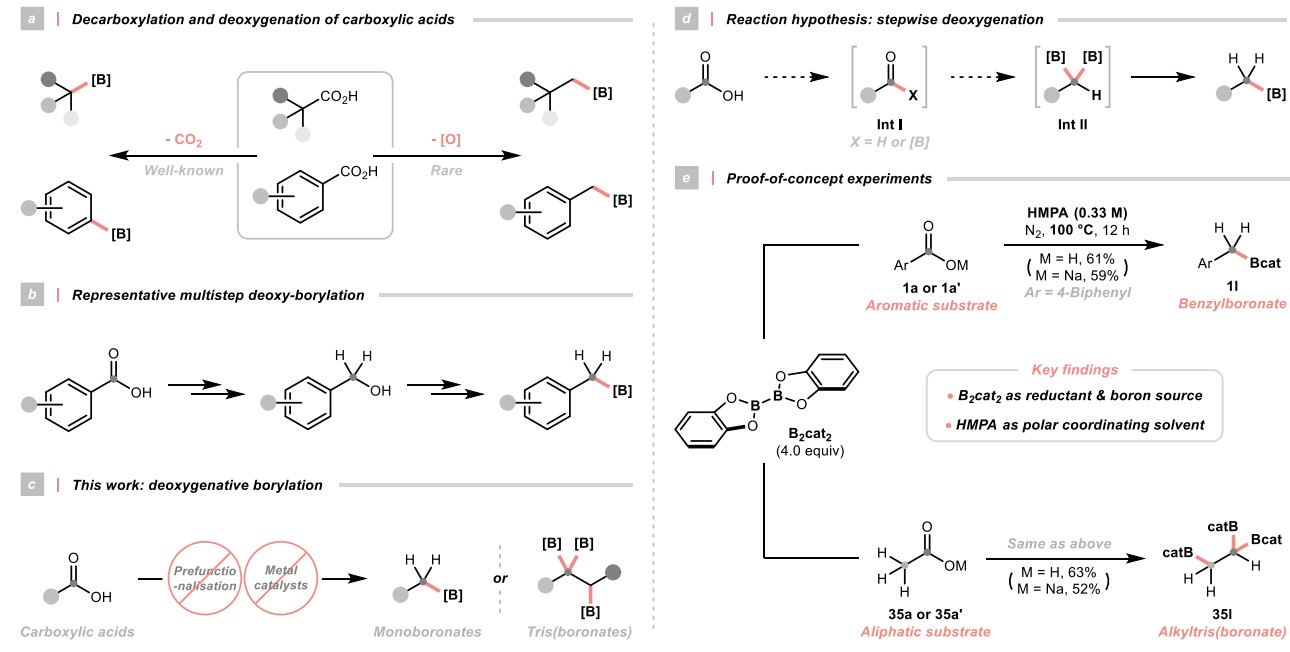

**Fig. 1 Introduction. a** Decarboxylation and deoxygenation of carboxylic acid; **b** Representative multistep deoxy-borylation; **c** This work: deoxygenative borylation; **d** Reaction hypothesis: stepwise deoxygenation; **e** Proof-of-concept experiments. For detailed conditions, see Supplementary Information. Yields were determined by NMR. [O], oxygenous functionality; [B], boron-containing functionality; TM transition metal, B₂cat₂ bis(catecholato)diboron, HMPA hexamethylphosphoramide.

divergent results with alkyl substrates, allowing access to several synthetically appealing but hard-to-access 1,1,2-tris(boronates) from the aliphatic acids.

## Results and discussion

**Evaluation of reaction conditions**. The key to enabling the desired deoxygenative reactivity is strategising the scissions of three thermodynamically and kinetically unreactive C–O bonds in the –$CO_2H$ group. For this purpose, we envisioned that a stepwise deoxygenation scenario could be operative (Fig. 1d). Hypothetically, the initial replacement of the carboxylic hydroxyl group with hydrogen or boron moiety could degrade the carboxylic acid into a more reactive carbonyl intermediate. Consecutive removals of C–O bonds through boron group transfer could give a *gem*-diboron intermediate, which was susceptible for protodeboronation to furnish the final product. Interestingly, similar acyl species have been proposed as intermediacy for carboxylic acid deoxygenative phosphorylation and carbon monoxide reduction[25–28].

Brief reaction analysis revealed that an oxygen transfer reagent and boron source were required for the expected deoxy-borylation. Recent efforts in our laboratory[29,30] and others'[31–40] suggested that diboron reagents, in concert with some Lewis bases[41], could serve both purposes. Specifically, some earlier discoveries from Aggarwal's[24,33,40], Glorius's[34], Studer's[35,39], Shi's[36] and our groups[30] unveiled a unique diboron reagent, bis(catecholato) diboron ($B_2cat_2$), that could effect the reductive borylation of various carbon electrophiles under metal-free conditions. Mindful of these successful examples, we considered harnessing $B_2cat_2$ for our long-sought carboxylic acid deoxygenative borylation project. Moreover, featuring superior electrophilicity and oxophilicity[42], $B_2cat_2$ could activate the acid substrates via coordination, forming boron oxides as innocuous by-products and providing the driving force to the process.

To establish proof-of-concept, 4-biphenycarboxylic acid (**1a**), acetic acid (**35a**) and their sodium salts (**1a'** and **35a'**) were selected as model substrates for aromatic and aliphatic substrates, respectively (Fig. 1e). Achieving the carboxylic acid deoxy-borylation needed considerable experimentation, which eventually led us to identify very simple and general conditions with 4.0 equiv $B_2cat_2$ in 0.33 M hexamethylphosphoramide (HMPA) at 100 °C as key parameters (see Supplementary Information). Control experiments showed that other diboron reagents, solvents and temperatures dramatically depressed the desired reactivity and the attempts with a wide range of additives did not give improved outcomes. These results collectively highlighted the unique effect of $B_2cat_2$ and HMPA in modulating deoxy-borylation. Noticeably, the successful synthesis of a small-molecule 1,1,2-tris(boronate) (**35b**) as a functionalisable ethylating reagent from the prevalent acetic acid represented significant advances since other plausible synthetic routes to access **35b** would involve multiple inconvenient steps with costly catalysts and gaseous materials[43–45].

**Substrate scope of deoxygenative borylation**. With the optimal conditions in hand, we explored the substrate scope of this methodology (Fig. 2). First, several typical boronic acid derivatives were prepared (**1 f** to **1k**) simply through transesterification with diols, diacid and diamine. Second, a wide spectrum of aromatic carboxylic acids featuring different electronic and steric profiles were evaluated. Benzoic acid and other derivatives bearing aliphatic (methyl, ethyl and butyl) and π-extended (phenyl, naphthyl) groups with varied substitution patterns were all effective substrates (**2b** to **13b**). To be noticed, the *ortho*-substituted benzoic acids with moderate steric hindrance could be

tolerated (**9b** and **12b**), among which the sodium carboxylates exhibited higher reactivities compared to the free acids. Some weakly electron-withdrawing or -donating (pseudo)halide functionalities (fluoro, bromo, chloro, iodo and acetoxy groups) were compatible with this reductive transformation, offering a variety of benzylboronates with two functional handles of orthogonal reactivities (**14b** to **20b**). Potentially, these building blocks could be applicable in iterative cross-couplings. Interestingly, a synthetically appealing bimetallic reagent (**21b**) could be easily obtained from the -Bpin appended benzoic acid (**21a**), which could be problematic in some metal catalyst-based conditions due to the facile transmetallation. To our delight, some electron-rich acids with alkoxy, hydroxy, amino and thioethereal substituents and the electron-poor one with –$OCF_3$ group could withstand our optimal conditions (**22b** to **31b**). In particular, those alkoxylated substrates, which were common in the benzoic acid family, could be converted to the desired products in high yields, irrespective of the competing coordination issue[30]. Moreover, the reactions of *N*-, *O*-, and *S*-heteroaromatic acids proceeded efficiently, granting several boronates with carbazole, benzofuran and benzothiophene cores (**32b** to **34b**).

Apart from aromatic examples, the performance of some representative aliphatic acids was examined. Our deoxytriboration protocol exhibited appreciable generality with these aliphatic compounds. Importantly, some light acids could undergo the α, α, β, -triboration exclusively, synthesising the low-weight 1,1,2-tris(boronates) that were, heretofore, difficult to prepare by conventional means (**35b** and **36b**)[46]. Other fatty acids, which were readily accessible from natural or commercial sources, could also be leveraged to prepare some longer-chain members in this valuable subset of organoboron (**37b** to **40b**). Due to its simplicity and generality, we anticipated that this method could become the routine synthetic route of 1,1,2-tris (boronates) and accelerate the in-depth study of their deborylative reactivities and application. Interestingly, simple formic acid and some hindered aliphatic congeners afforded *gem*-diboronates as exclusive deoxygenation products under the same conditions (**41b'** to **44b'**).

**Synthetic utility**. To showcase the synthetic utility of this strategy, several applications were conceived. In probing the robustness of our conditions, carboxylic acids derived from the natural products, drug molecules and others with complex scaffolds were subject to our deoxygenative borylation (Fig. 3a). Encouragingly, all of them could be borylated smoothly, giving boronates with diverse molecular complexity (**45 g**, **46b** to **51b**). In particular, benzylboronate **45 g** with a bulky and coordinating phosphino substituent could be useful in designing some phosphine ligands and catalysts; however, its synthesis could cause problems in traditional metal-involved approaches, emphasising the advantages of our metal-free strategy.

Given the low cost of some isotope-containing carboxylic acids, they were submitted to our deoxygenation conditions, resulting in the labelled boronates in good yields (Fig. 3a). These building blocks could be further diversified, expediting a modular strategy for $^2$H- and $^{13}$C-labeled compounds synthesis. Partnered with many elegant methods to attain carboxylic acids, several one-pot, two-step transformations could be devised using other attractive starting materials to make boronates. To our delight, we discovered that this method was capable of converting the mixed aldehydes and carboxylic acids, regardless of their ratios, to the same boronates (Fig. 3b, left). Taking advantage of this prolific conversion, alcohol and methylarene oxidation were successfully fused in our borylation protocol, providing the relevant boronates (**1b** and **39b**) without isolating the oxidation intermediates (Fig. 3b, right). Moreover, by

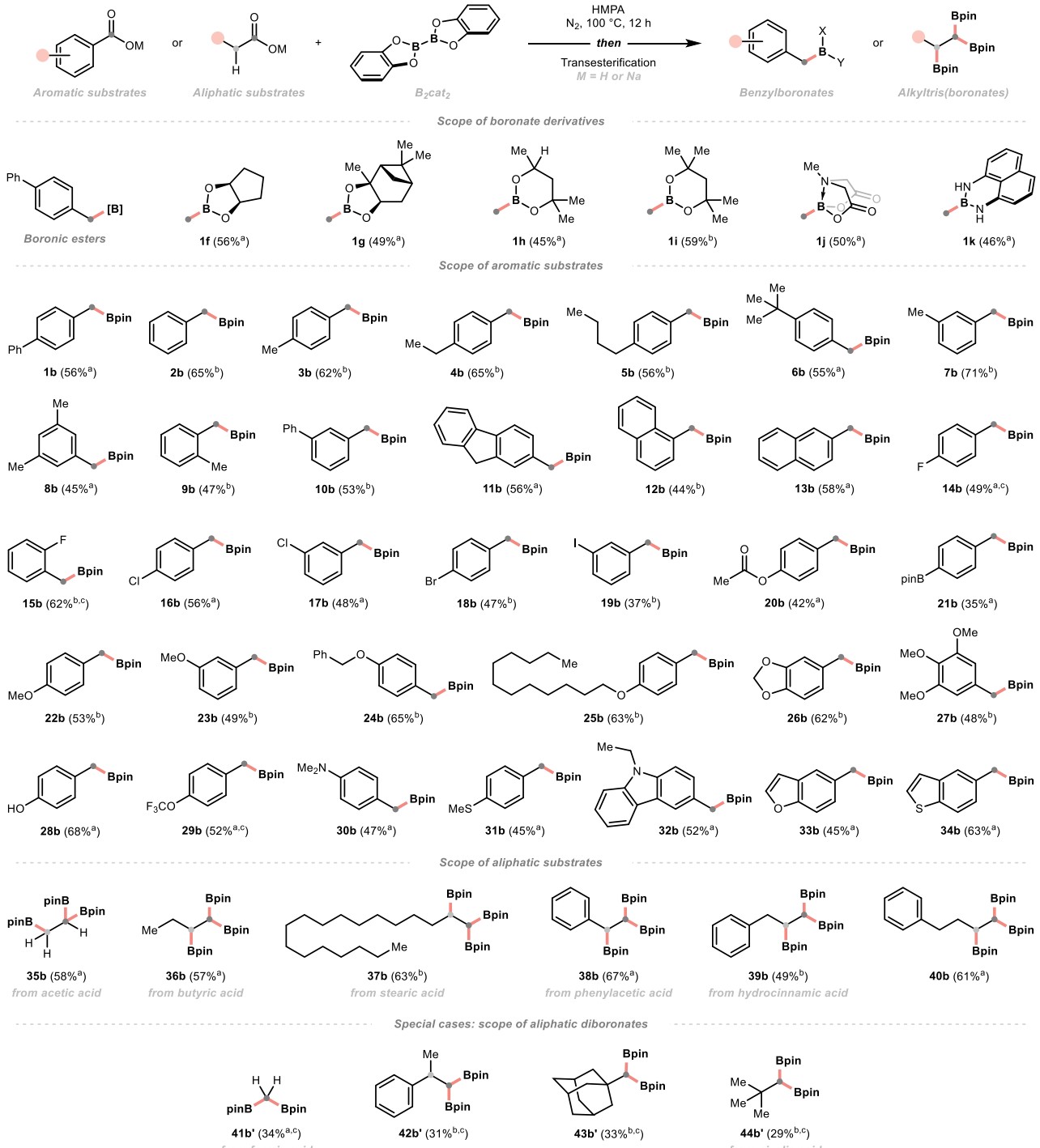

**Fig. 2 Substrate scope of deoxygenative borylation.** For detailed conditions, see Supplementary Information. Yields are for isolated products after purification unless otherwise specified. [a]Condition A: reactions were performed with free carboxylic acid (0.20 mmol, 1.0 equiv), $B_2cat_2$ (0.80 mmol, 4.0 equiv) in 0.33 M HMPA under nitrogen at 100 °C for 12 h; then transesterification. [b]Condition B: same as condition A but performed with sodium carboxylates. [c]Reaction yields were determined by NMR due to the difficulties of isolation (e.g. volatility or instability of targeted compounds). Bpin boronic acid pinacol ester.

merging the carboxylation and borylation, we could orchestrate a concise strategy to prepare the benzylboronate from Grignard reagent, with $CO_2$ as a C1 synthon (Fig. 3c). Notably, simply by iterating the deoxygenative borylation and Suzuki-Miyaura coupling, several readily available oxygenated chemicals could be stitched, affording a key intermediate for paracyclophane synthesis as well as a series of pharmaceutically relevant compounds with diarylmethane scaffolds, including *O*-methyltransferase (OMT)

inhibitor precursors and drug candidates for arenavirus treatment (Fig. 3d)[47].

**Mechanistic considerations.** To gain insights into this deoxygenative borylation reaction, a series of experiments and computer modelling were conducted (see Supplementary Information). Preliminary exploration involving radical scavengers and carbene

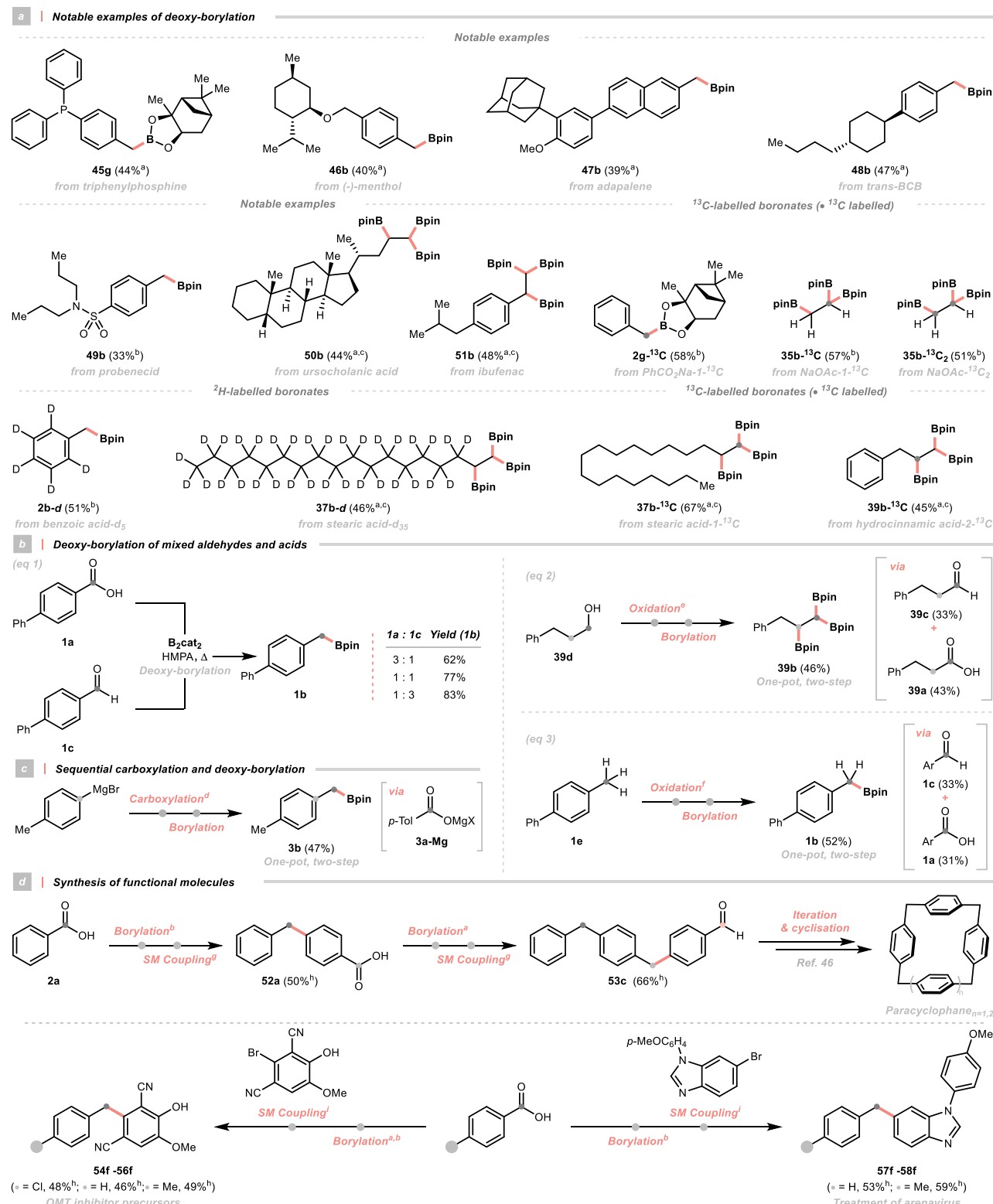

**Fig. 3 Synthetic applications. a** Notable examples of deoxy-borylation; **b** Deoxy-borylation of mixed aldehydes and acids; **c** Sequential carboxylation and deoxy-borylation; **d** Synthesis of functional molecules. For detailed conditions, see Supplementary Information. [a]Refer to Condition A in Fig. 2. [b]Refer to condition B in Fig. 2. [c]Refer to condition A in Fig. 2 but in DMA. [d]p-TolMgBr in Et$_2$O under CO$_2$ atmosphere (1.0 atm) at from 0 °C to rt for 1.0 h. [e]Alcohol, Fe(NO$_3$)$_3$·9H$_2$O, (2,2,6,6-tetramethylpiperidin-1-yl)oxyl (TEMPO), KCl in DCE under air at rt for 8.0 h. [f]NaSO$_2$CF$_3$ in CH$_3$CN under O$_2$ atmosphere (1.0 atm) with visible light at rt for 12 h. [g]R-BF$_3$K, Ar-Br, Pd(dppf)Cl$_2$·CH$_2$Cl$_2$, Cs$_2$CO$_3$ in THF/H$_2$O under air and reflux for 24 h. [h]Yields refer to SM coupling. [i]R-Bpin, Ar-Br, Pd(dppf)Cl$_2$·CH$_2$Cl$_2$, NaHCO$_3$ in EtOH/H$_2$O under air and reflux for 3.0 h. Bpin boronic acid pinacol ester, Bpnd boronic acid (+)−pinanediol ester, BCB 4-butylcyclohexylbenzoic acid, p-Tol p-tolyl, SM coupling Suzuki-Miyaura coupling, OMT O-methyltransferase, rt room temperature.

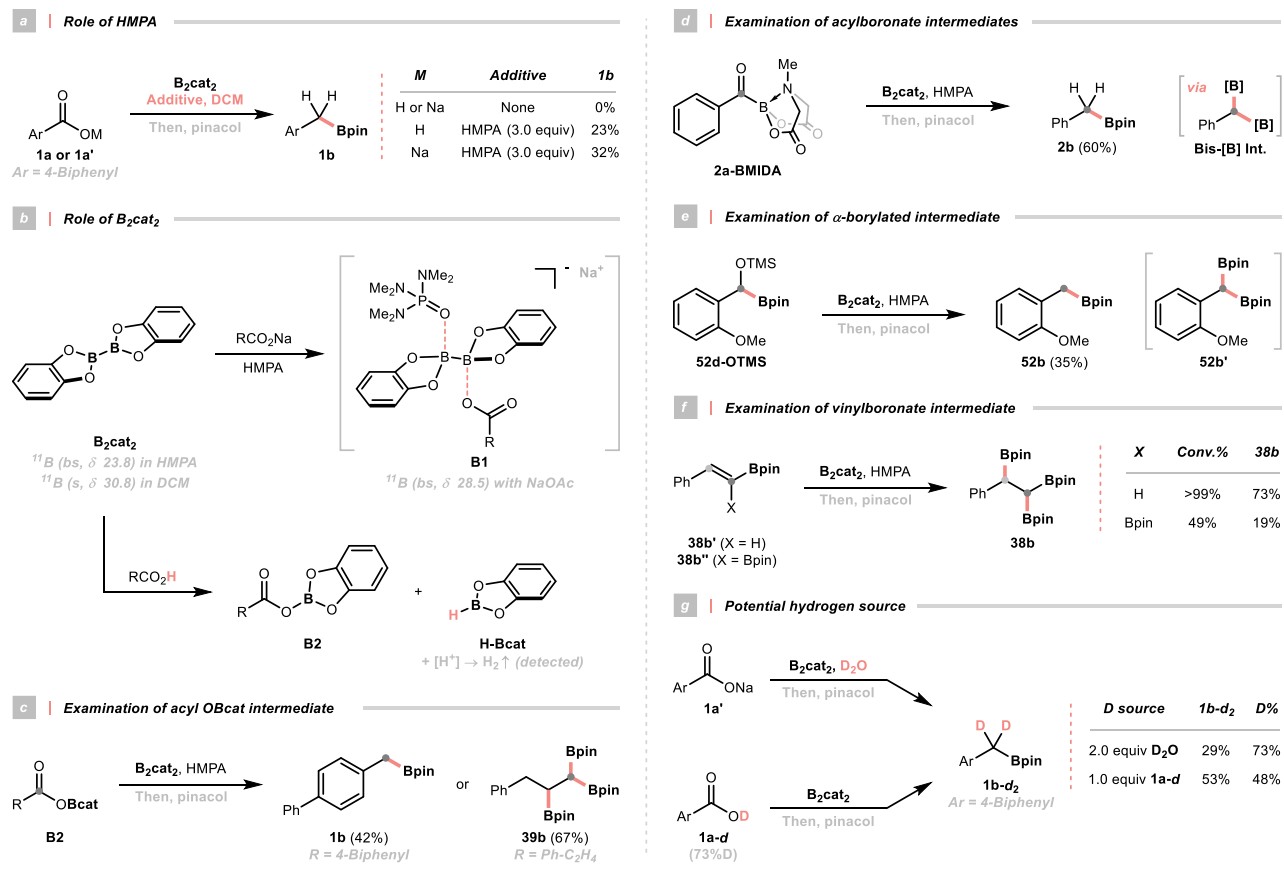

**Fig. 4 Key mechanistic discoveries.** For detailed conditions, see Supplementary Information including **a** Role of HMPA; **b** Role of B$_2$cat$_2$; **c** Examination of acyl OBcat intermediate; **d** Examination of acylboronate intermediate; **e** Examination of α-borylated intermediate; **f** Examination of vinylboronate intermediate; **g** Potential hydrogen source. Unless otherwise specified, the deoxygenative borylation were performed under optimal conditions and yields were determined by NMR. DCM dichloromethane, BMIDA boronic acid methyliminodiacetic ester, TMS trimethylsilyl.

precursor[48] implied that the radical and carbene mechanism was less likely.

A pronounced solvent effect was observed as amide-type solvents were mandatory for this transformation, signifying the importance of a mildly basic and coordinating environment. Intriguingly, the deoxy-borylation did not proceed in dichloromethane (DCM, a non-coordinating control solvent); yet, the targeted product could be retrieved by adding slightly excessive HMPA (Fig. 4a). Considering the Lewis acidity of B$_2$cat$_2$, its boron centres could be ligated with the basic oxygen of HMPA. In line with this surmise, an upfield and broadened $^{11}$B nuclear magnetic resonance (NMR) signal was seen in HMPA (23.8 ppm) relative to DCM (30.8 ppm), substantiating a loose complexation of B$_2$cat$_2$ with the former solvent. Such an acid-base interaction could also be visualised in $^{1}$H and $^{31}$P NMR. As expected, comparable $^{11}$B chemical shift was observed by mixing B$_2$cat$_2$ and NaOAc, revealing a similar ligation mode between HMPA and sodium carboxylate. Consequently, a ternary species **B1** was postulated, in which an HMPA and carboxylate molecule individually resided on boron centres of B$_2$cat$_2$ (Fig. 4b). Consistent with some precedential examples[23,24,30,33,34,40,49], the formation of such a boron aggregate was evidenced by spectroscopic techniques, where a distinct ultraviolet-visible (UV-vis) absorption behaviour of the diboron and carboxylate mixture in HMPA was shown compared to the individual components. Conversely, the combination of B$_2$cat$_2$ and PhCO$_2$H in HMPA remained mostly unchanged in the UV-vis spectrum, suggesting a dissimilar pathway in the cases of free acids. Being aware of the reductibility of diboron reagents[50,51], B$_2$cat$_2$

might reduce carboxylic acid into the mixed anhydride **B2**, with concurrent formation of borohydride (H-Bcat). Unsurprisingly, hydrogen (H$_2$) evolution was detected upon quenching the reaction with MeOH. Independently prepared aromatic and aliphatic OBcat anhydrides were also proven as active intermediates, giving the corresponding boronates (**1b** and **39b**) in good yields (Fig. 4c).

Next, the mechanism of breaking the C–O bonds was studied. In accordance with our stepwise deoxygenation design and some previous work[30,52–54], several oxygenous compounds were synthesised and subjected to standard conditions for deoxygenative borylation. Among them, **2a-BMIDA** and **52d-OTMS**[55,56], conserving a C=O and C–O bond, respectively, were identified as competent intermediates (Fig. 4d and e). Presumably, *gem*-diboronates could be involved during the reduction of the above-mentioned species[48,57], which was validated by the selective mono-protodeboronation of α, α-benzyldiboronate **2b'** under our optimal conditions.

Regarding the reaction pathway for aliphatic acid triboration, analogous acylboronate and α-borylated alcoholate could be engaged, the latter of which might lead to a vinylboronate intermediate via elimination of the hydroxyborate[58,59]. Accordingly, monoboronate **38b'** and diboronate **38b''** were submitted to the triboration conditions. Pleasingly, triboronate **38b** was yielded in both cases (Fig. 4f). The higher conversion and productivity of **38b'** illustrated that it might be the major contributor to the triboronate formation. It was worth noting that both vinylboronate and vinyldiboronate were identified in reaction crude of encumbered aliphatic acid (**42a**).

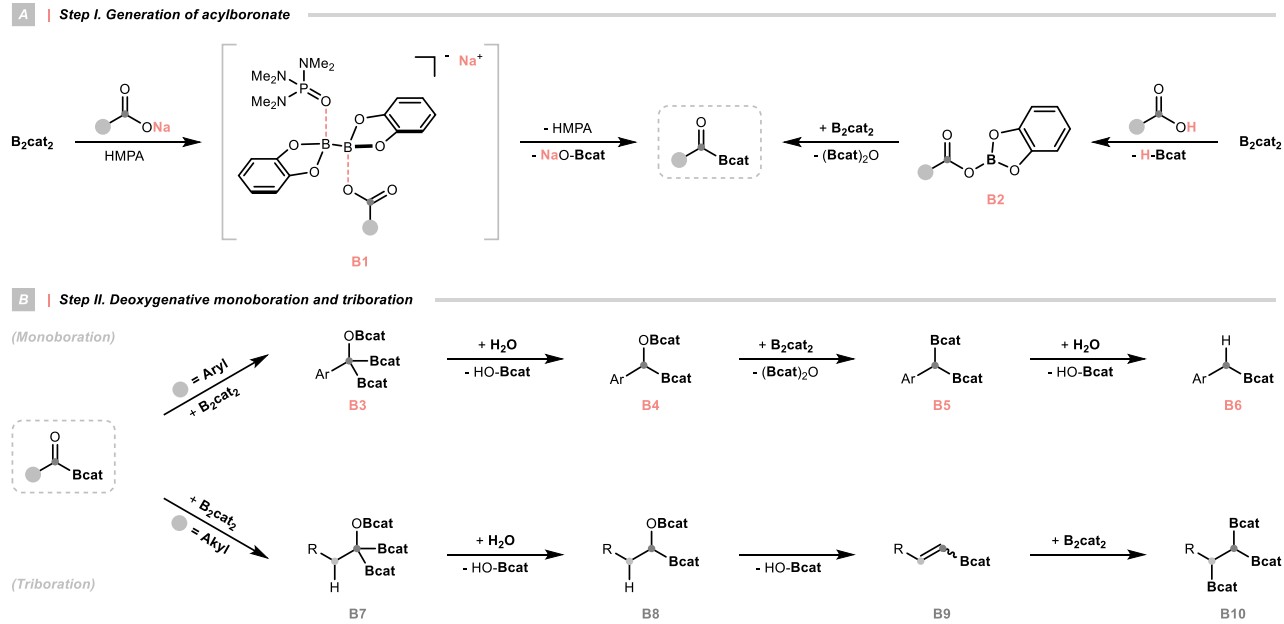

**Fig. 5 Proposed mechanism. A** Step I. Generation of acylboronate; **B** Step II. Deoxygenative monoboration and triboration. $B_2cat_2$ bis(catecholato)diboron, Bcat boronic acid catechol ester.

Finally, isotope-labelling experiments were performed, unambiguously determining that the benzylic hydrogen of **1b** could stem from either water residue or carboxylic proton (Fig. 4g). It was also noteworthy that alcohols and their *O*-borates remained unreactive under the optimal conditions, and the potential intermediacy of aldehydes and carboxylic acid anhydrides could not be fully excluded.

Taken together, a tentative mechanism was depicted (Fig. 5). In HMPA, $B_2cat_2$ could coordinate with $RCO_2Na$ to form **B1**, simultaneously activating the diboron and carboxylate. Then, an HMPA-mediated boron-oxygen exchange occurred, giving an acylboronate by losing the sodium borate (Fig. 5A left). The same acylboron could result from a borylative displacement with the mixed anhydride **B2**, which was derived from the redox reaction between $RCO_2H$ and $B_2cat_2$ (Fig. 5A right). For the aromatic substrates, sequential diborylation and protodeboronation led to the vicinal *C,O*-bisborylated **B4**, which could undergo a C–O borylation via the metathesis with another $B_2cat_2$. The resulting *gem*-diboron **B5** could be selectively hydrolysed into the final benzylboronate product **B6** (Fig. 5B upper). In terms of aliphatic acids, after the bisborylation and hydrolysis, elimination occurred preferentially, which was followed by another bisborylation to deliver the tris(boronate) **B10** (Fig. 5B lower).

In summary, we have successfully established a concise and general deoxygenative borylation tactic to access a series of alkylboronates from carboxylic acids under mild conditions, which was distinct from its well-developed decarboxylative counterpart. The simple but unique combination of commercially available $B_2cat_2$ and HMPA constitutes this straightforward strategy, enabling the synthesis of various benzylboronate and alkyl (trisboronates) from aromatic and aliphatic substrates, respectively. In this method, we strategised the challenging cleavage of three C–O bonds of the carboxylic group in a stepwise manner and tactically eliminated the involvement of metal catalysts and any preactivation steps. Detailed mechanistic studies revealed a common acyl intermediate, which could be further deoxygenated and gave divergent borylation outcomes depending on the substrates' nature. In light of their simplicity and versatility, we envisaged that the established reductive borylation methods will

interest, inspire and propel more researchers for their future efforts in the exciting area of deoxygenative transformations and continuously contribute to such evolving chemistry.

## Methods

**General procedure for deoxygenative borylation of carboxylic acid and their sodium salts.** The preparation of **1b** from **1a** (M=H) or **1a'** (M=Na) is applicable to all boronic acid pinacol ester synthesis in this work unless otherwise specified.

To a flame-dried reaction tube (10 mL) equipped with a Teflon-coated magnetic stirring bar were added 4-biphenylcarboxylic acid (**1a**, 39.6 mg, 0.20 mmol, 1.0 equiv) or 4-biphenylcarboxylic acid sodium salt (**1a'**, 44.0 mg, 0.20 mmol, 1.0 equiv) and $B_2cat_2$ (190.2 mg, 0.80 mmol, 4.0 equiv). The reaction tube was covered by a rubber septum, and the resulting mixture was degassed and back-filled with nitrogen ($N_2$, high purity 4.8, >99.998%) before being transferred to the nitrogen-filled glovebox. Shortly after, 0.60 mL HMPA (0.33 M) was syringed into the reaction tube, which was capped by an aluminium seal with PTFE/silicone septum. The reaction tube was moved out of the glovebox and stirred at 100 °C for 12 h.

Upon the completion of the reaction, to the resulted crude was added pinacol (189.1 mg, 1.6 mmol, 8.0 equiv) and 0.70 mL $Et_3N$. Then, the mixture was stirred at ambient temperature for 1.0 h, which was quenched by 3.0 mL brine, 1.0 mL distilled $H_2O$ and supplemented with 5.0 mL EtOAc for extraction. With vigorous shaking followed by unperturbed standing for a while to allow the two layers to separate, the upper organic layer was transferred and passed through a short-packed pipette column filled with $Na_2SO_4$ (3.0 cm) and silica gel (0.5 cm). The above-mentioned extraction process was repeated another four times with EtOAc (4*5 mL). The filtered anhydrous organic solution (~25 mL) was collected in a 100 mL round bottom flask and concentrated on a rotary evaporator. The crude was subjected to flash column chromatography on silica gel to furnish the titled compound **1b**.

## Data availability

The authors declare that all other data supporting the findings of this study are available within the article and Supplementary Information files, and are available from the corresponding author upon reasonable request.

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

## Acknowledgements

We are grateful to the Canada Research Chair Foundation (to C.-J.L.), the Canada Foundation for Innovation, the FQRNT Center in Green Chemistry and Catalysis, the Natural Sciences and Engineering Research Council of Canada and McGill University for supporting our research. We would like to acknowledge the McGill Chemistry Characterization Facility for their contribution to the compound characterisation in this work, to be specific, Robin Stein on the NMR, Nadim Saadé and Alexander Wahba on HRMS. We are indebted to Ranjan Roy

(McGill University) for our access to the spectroscopic facilities. We would also like to thank our group members and colleagues for their generous help in polishing the manuscript.

## Author contributions

J.L. conceived, designed and optimised the reaction. J.L. and C.Y.H. conducted the experiments. M.A. performed the calculation. The manuscript was written through the contributions of all authors. All authors have given approval to the final version of the manuscript.

## Competing interests

The authors declare no competing interests.
