## [Peer Review File · Nature Communications]

Reviewers' Comments:

Reviewer #1:

Remarks to the Author:

Li and coworkers report a direct deoxygenative borylation of free carboxylic acids using B2Cat2. It is a novel and elegant paper that develops a simple, practical, and useful protocol for achieving mono- and tri-borylated compounds. The authors demonstrate good functional group compatibility (>40 examples) in reasonable yields. The diversifications of products also show the potential of the current methods in synthetic chemistry. The mechanistic studies rationalize the product outcome of the reaction. I believe this work will be well-received by the community. Overall, this paper is well written, and the work is thorough, and publication in Nature Communications is strongly recommended after addressing the following issues:

1. In SI, ¹¹B NMR spectra should be included in addition to ¹H and ¹³C NMR.
2. How about α -branched carboxylic acid? For example, α -methyl phenylacetic acid.
3. Did the authors try pivalic acid? Logically, the absence of an α -proton could avoid elimination therefore a mono-borylated product would be formed.
4. Can this method be compatible with pyridine? For example, 2-, or 3-, or 4-pyridine carboxylic acid?
5. I notice that this method is compatible with the free hydroxy group, thus I recommend the authors try more bioactive substrates cholic acid, and deoxycholic acid, which should add value to the current work.

Reviewer #2:

Remarks to the Author:

The manuscript from Li and co-workers describes a deoxygenative borylation reaction, which takes advantage of carboxylic acids and B2cat2 as substrates and is assisted by HMPA. Although the same group has disclosed the deoxygenative borylation reaction of aldehyde (J. Am. Chem. Soc. 2020, 142, 30, 13011) recently, the novelty of this work is also characterized by its carboxylic starting materials. Generally, this manuscript is well written and the provided data can well support the conclusion. This work should be not only a supplement to the existing methods that convert acids to boronic derivatives but also a creative one-carbon homologation strategy. Also, this work is featured by the diverse transformation that takes best use of this deoxygenative borylation reaction. The provided abundant mechanism information gives a detailed reaction pathway for this reaction. In view of this, the reviewer recommends publication of this manuscript in Nature Communications.

Before acceptance of this manuscript, several issues that are mentioned below need further discussion:

1. In the substrate scope of deoxygenative borylation of aromatic carboxylic acids, it seems only the benzylic position of phenyl ring is reactive? Has any other heterocyclic substrate been tried?
2. Has any amino acid been used as starting material?
3. In Scheme 4d, acyl BMIDA intermediate is used to probe the reaction mechanism. After treating with pinacol, the BPIN product 1b is finally obtained. Which type of boronic species could be obtained before this treatment? BMIDA or BCAT species?
4. In supporting information, the authors noted "all the boronates were stored at -20 °C to prevent significant decomposition." Since the products are isolated after repeated extraction using water, they should not be very sensitive towards hydrolysis. Could some information about such decomposition given? Should hydrolysis or oxidation be prevented while storing such compounds?
5. In Table S5.1.1, should "borylating reagent" be "additive"?

Reviewer #3:

Remarks to the Author:

This study report an interesting reaction that can transform carboxylic acid into benzylic boron compounds or alkytrisboron compounds with B2Cat2 without transition metal catalysts. This provides a new transformation pattern for the production of organoboron compounds. But the

author already reported a reaction of B2cat2, which gives similar products from aldehydes (ref. 25, JACS 2020, 13011). Thus the present reaction is the carboxylic acid version of the previous JACS paper. Besides, this reaction requires a large excess of HMPA, which is a strong carcinogen, as a solvent and relatively high temperature (100 degrees C). The author mentions the advantage of metal-free conditions, but the reaction condition is a toxic and not environmentally benign process. From these points, it is difficult for me to see this paper has enough novelty and usefulness suitable for Nat. Comm.

Other points:

1. The reaction mechanism is still not clear. If the proposed reaction mechanism supposed by the authors can be employed (in Scheme5A), the most important key step is B1 or B2 intermediate to acyl boron compound. This can be investigated by DFT calculations.

2. I don't feel several application studies in Scheme 3 were not variable and appealing. For Scheme 2B, not many researchers want to convert benzoic acid to 1-phenylacetic acid by the present procedure (B2cat2/HMPA; Cu cat/CO₂). Similarly, preparations of 2b and 38b'-H₂ are little to be gained for the effort. For making these processes more appealing, reactions including more complicated starting materials with functional groups to hard-to-obtain-products should be shown.

Reviewer 1

Li and co-workers report a direct deoxygenative borylation of free carboxylic acids using B_2cat_2 . It is a novel and elegant paper that develops a simple, practical, and useful protocol for achieving mono- and tri-borylated compounds. The authors demonstrate good functional group compatibility (>40 examples) in reasonable yields. The diversifications of products also show the potential of the current methods in synthetic chemistry. The mechanistic studies rationalise the product outcome of the reaction. I believe this work will be well-received by the community.

Response: Thank you for your comment on our work.

Overall, this paper is well written, and the work is thorough, and publication in Nature Communications is strongly recommended after addressing the following issues:

1. In SI, ^{11}B NMR spectra should be included in addition to 1H and ^{13}C NMR.

Response: The ^{11}B spectra of all boron-containing compounds have been added in the supplementary information for full characterisation.

2. How about α -branched carboxylic acid? For example, α -methyl phenylacetic acid.

Response: To study the reactivities of α -branched carboxylic acids, several representative members in this class, including 2-phenylpropionic acid, 2-methylbutanoic acid, ibuprofen, and naproxen, were utilised. We carefully analysed the reaction mixture in the case of 2-phenylpropionic acid using NMR and HRMS. The results were shown below.

Although the desired tris(boronate) product was not observed in this reaction, several boron-containing products were identified.

1) 1,1-Alkyldiboronate: this side product could be attributed to the selective monoprotonation of desired tris(boronate) product. We believed that the strained scaffold of tris(boronate) product promoted such a hydrolysis step during the reaction since it occurred to a much smaller extent in the case of linear acid examples. In addition, the internal chelation between Bpin groups was proposed to assist the deborylation step, where the carbanion bearing α -phenyl ring was favoured (*J. Am. Chem. Soc.* **136**, 16140-16143 (2014)).

2) Vinylboronate: Due in part to the steric hindrance of this branched substrate, the rate of the vicinal diborylation step was decreased, allowing the accumulation of vinylboronate and 1,1-vinyldiboronate in the reaction mixture. The formation of these intermediates was consistent with our mechanistic findings, which indicates the important role of vinyl(di)boronates in the deoxygenative borylation of aliphatic substrates.

Similar results were obtained in the cases of 2-methylbutanoic acid, ibuprofen, and naproxen (not shown above).

3. Did the authors try pivalic acid? Logically, the absence of an α -proton could avoid elimination; therefore, a mono-borylated product would be formed.

Response: We have tested the reactivity of some carboxylic acids bearing α -tertiary carbon centres, e.g., pivalic acid and 1-adamantanecarboxylic acid. The results were shown in the scheme below.

As mentioned by the reviewer, the absence of α -protons could avoid elimination; therefore, the formation of 1,1,2-triboronate was impossible with these substrates. However, unlike the aromatic acids (giving benzylmonoboronates), diboronates were obtained as exclusive products in these cases. This result was, indeed, consistent with our previous findings that steric hindrance would decelerate the protodeboration (*J. Am. Chem. Soc.* **142**, 13011-13020 (2020)). Other than the steric factors, the absence of resonance stabilisation of the anionic intermediate may account for the slow hydrolysis rate, leading to exclusive formation of the *gem*-diboronate in these cases.

Besides, we noticed the low yields in these examples, which could be attributed to the encumbered structures of these acids, disfavoring the complexation between acid and diboron.

4. Can this method be compatible with pyridine? For example, 2-, or 3-, or 4-pyridine carboxylic acid?

Response: We have tested the reactivities of the pyridinecarboxylic acid series, including picolinic acid, nicotinic acid and isonicotinic acid. The results were shown in the scheme below.

All three isomers of the pyridinecarboxylic acids did not give the corresponding pyridylic boronates under our optimal conditions, whether in the free acid or sodium salt forms. This was in accordance with our results in testing the effect of basic additives, in which pyridine showed the inhibitory effect.

Possible rationales for these poor reactivities include:

1) Pyridine shows strong coordination to the diboron compound, which was a typical interaction between Lewis acidic boron compound and the nitrogen base. Characteristic yellow to red colour was observed when mixing the pyridine-based substrates with B₂cat₂. For selected references, see *J. Am. Chem. Soc.* **139**, 7440-7443 (2017).; *Org. Lett.* **19**, 4291-4294 (2017).;

2) Some pyridine-diboron complexes could induce dearomatization of the pyridine ring, which could consume the B₂cat₂ in our case and hamper the desired reactivity. For selected references, see *Chem. Sci.* **9**, 2711-2722 (2018).; *Org. Lett.* **21**, 9812-9817 (2019).; *J. Am. Chem. Soc.* **141**, 9124-9128 (2019).; *Chem. Sci.* **11**, 742-747 (2020).; *Angew. Chem., Int. Ed.* **59**, 2095-2099 (2020).; *J. Org. Chem.* **86**, 3287-3299 (2021)..

5. I notice that this method is compatible with the free hydroxy group. Thus, I recommend the authors try more bioactive substrates cholic acid, and deoxycholic acid, which should add value to the current work.

Response: The reactivities of several bile acids and their derivatives were evaluated under our optimal conditions. The results were shown below.

Among the bile acids tested, although both the cholic acid and deoxycholic acid cannot afford the desired borylated product, 5 β -cholic acid was successfully converted to the corresponding tris(boronate).

To explain such different reactivities, several control experiments were conducted. Under the same conditions, 4-hydroxybenzoic acid was deoxygenatively borylated, while the 4-(hydroxymethyl)benzoic acid did not impart any desired reactivity. These results indicated that our system could tolerate the phenolic proton but not the alcoholic one since the latter, which is more nucleophilic, might transesterify with B₂cat₂, generating the B₂pin₂-like diboron. This new alcohol-bound diboron was shown inactive since B₂pin₂ was unable to perform the deoxygenative borylation, neither with the free acid nor the sodium carboxylate.

In addition to the bile acid series, we also supplemented several structurally complex samples in the revised manuscript, including some pharmaceutically relevant and isotope-labelled (²H and ¹³C) carboxylic acids.

Notable examples

Notable examples

¹³C-labelled boronates (* ¹³C labelled)

²H-labelled boronates

¹³C-labelled boronates (* ¹³C labelled)

Reviewer 2

The manuscript from Li and co-workers describes a deoxygenative borylation reaction, which takes advantage of carboxylic acids and B_2cat_2 as substrates and is assisted by HMPA. Although the same group has disclosed the deoxygenative borylation reaction of aldehyde (*J. Am. Chem. Soc.* **2020**, *142*, 30, 13011) recently, the novelty of this work is also characterised by its carboxylic starting materials. Generally, this manuscript is well written, and the provided data can well support the conclusion. This work should be not only a supplement to the existing methods that convert acids to boronic derivatives but also a creative one-carbon homologation strategy. Also, this work is featured by the diverse transformation that takes the best use of this deoxygenative borylation reaction. The provided abundant mechanism information gives a detailed reaction pathway for this reaction. In view of this, the reviewer recommends the publication of this manuscript in *Nature Communications*.

Response: Thank you for your comment on our work.

Before acceptance of this manuscript, several issues that are mentioned below need further discussion:

1. In the substrate scope of deoxygenative borylation of aromatic carboxylic acids, it seems only the benzylic position of the phenyl ring is reactive? Has any other heterocyclic substrate been tried?

Response: Several representative heteroaromatic carboxylic acids/carboxylates with the carboxyl/carboxylate groups on heteroaromatic rings were subjected to our optimal conditions. The results were shown in the scheme below.

In general, the carboxylic/carboxylate groups on heteroaromatic rings showed lower reactivity compared to benzoic acid and its derivatives. This is possibly caused by the intrinsic instability of heteroaromatic rings, which could undergo hydrolytic ring opening or reductive dearomatisation under our conditions (B_2cat_2 could behave as strong Lewis acid and reductant). For those with five-membered heterocyclic moieties, their electron-rich carboxyl/carboxylate groups were more reluctant toward reduction.

To be noticed, all three isomers of the pyridinecarboxylic acids did not give the corresponding pyridylic boronates under our optimal conditions, whether in the free acid or sodium salt forms. This

was in accordance with our results in testing the effect of basic additives, in which pyridine showed an inhibitory effect even in a catalytic amount.

Possible rationales for these poor reactivities include:

1) Pyridine shows strong coordination to the diboron compound, which was a typical interaction between Lewis acidic boron compound and the nitrogen base. Characteristic yellow to red colour was observed when mixing the pyridine-based substrates with B_2cat_2 . For selected references, see *J. Am. Chem. Soc.* **139**, 7440-7443 (2017).; *Org. Lett.* **19**, 4291-4294 (2017).;

2) Some pyridine-diboron complexes could induce dearomatisation of the pyridine ring, which could consume the B_2cat_2 in our case and hamper the desired reactivity. For selected references, see *Chem. Sci.* **9**, 2711-2722 (2018).; *Org. Lett.* **21**, 9812-9817 (2019).; *J. Am. Chem. Soc.* **141**, 9124-9128 (2019).; *Chem. Sci.* **11**, 742-747 (2020).; *Angew. Chem., Int. Ed.* **59**, 2095-2099 (2020).; *J. Org. Chem.* **86**, 3287-3299 (2021)..

We have included these samples and the corresponding description in the supplementary information.

2. Has any amino acid been used as starting material?

Response: Amino acids are valuable starting materials in organic synthesis. Considering the broad availability of glycine and its derivative, some unprotected and *N*-protected glycines were submitted to our deoxygenative borylation conditions. The results were summarised below.

Although the amino group should, in principle, be compatible with our borylation conditions (see the example of 4-*N,N*-dimethylaminobenzoic acid), the desired boron products from glycines with or with various protecting groups were not observed in NMR and GC-MS analysis. We suspected that the basic and coordinating amino group adjacent to the carboxylic group would interfere with the complexation between the B_2cat_2 and the latter, inhibiting the desirable reactivities and remaining as one of the limitations of our methods.

3. In Scheme 4d, acyl BMIDA intermediate is used to probe the reaction mechanism. After treating with pinacol, the Bpin product **2b** is finally obtained. Which type of boronic species could be obtained before this treatment? BMIDA or Bcat species?

Response: The acylboron compound **2a-BMIDA** was chosen in this mechanistic study because it is one of the few acylboronates that were fairly stable and readily accessible (*Angew. Chem., Int. Ed.* **59**, 16847-16858 (2020)).

To identify the active boron species before transesterification, we carefully analysed the reaction mixture before transesterification workup, and we did not observe the Bn-BMIDA **2j** in both NMR and GC-MS.

To determine the stability of **2j** under transesterification conditions, we performed the following experiments.

In both cases, the **2j** remained intact during the transesterification with pinacol. Most of the MIDA ester was recovered, and the pinacol ester was absent or present in trace amount.

Taken together, we concluded that the active boron species in the reaction crude of deoxygenation of acylboron is R-Bcat (due to its instability, isolation were unsuccessful).

4. In supporting information, the authors noted "all the boronates were stored at -20 °C to prevent significant decomposition." Since the products are isolated after repeated extraction using water, they should not be very sensitive towards hydrolysis. Could some information about such decomposition given? Should hydrolysis or oxidation be prevented while storing such compounds?

Response: We added this note to the supplementary information by referring to the storage conditions recommended by Combi Blocks. For all the benzylboronic pinacol esters sold by this

vendor, it mentioned, “Store under -20 °C or -40 °C if to be stored for more than 3 months. Keep the container tightly closed in a dry and well-ventilated place. Containers which are opened must be carefully resealed and kept upright to prevent leakage” (4-Methylbenzylboronic acid pinacol ester as a representative example, please see <https://www.combi-blocks.com/cgi-bin/find.cgi?PN-8008>).

To provide more information about handling and storing this type of compounds, we examined their sensitivity towards light, air and moisture using benzylboronic acid pinacol ester. The results were shown below.

compound	storage conditions at room temperature	recovery of 2b	other comments
 2b (0.20 mmol)	under nitrogen with ambient light 30 days	91%	2% PhCHO 1% PhCH ₂ OH 0% PhCH ₃
	under air with ambient light 30 days	88%	trace PhCHO 1% PhCH ₂ OH 0% PhCH ₃
	under nitrogen in the dark 30 days	96%	trace PhCHO 1% PhCH ₂ OH 0% PhCH ₃
	under air in the dark 30 days	80%	trace PhCHO 1% PhCH ₂ OH 0% PhCH ₃
	in 0.60 mL water under air with ambient light 12 h	95%	trace PhCHO 3% PhCH ₂ OH 0% PhCH ₃

In all these stability test experiments, a high recovery of **2b** was obtained, and an insignificant amount of decomposition products, including oxidation and hydrolysis, was observed; therefore, at least in the timeframe from half a day to a month, we could conclude that **2b** was not susceptible towards ambient light and atmosphere (O₂, H₂O, etc.).

To be noticed, we observed a relatively lower recovery rate when the samples were exposed to the air, which might attribute to the evaporation over time. Besides, due to the limited time, other functionalised benzylboronic acid pinacol esters were not tested, and the stability of **2b** should only serve as a standard for a small part of the members in the boronate family. Therefore, following the procedure of Combi Blocks to properly store these kinds of compounds are highly advised for long term storage.

5. In Table S5.1.1, should “borylating reagent” be “additive”?

Response: Thank you for your suggestion. We have corrected this mistake by replacing “borylating reagent” with “radical tapper”.

Reviewer 3

This study report an interesting reaction that can transform carboxylic acid into benzylic boron compounds or alkytrisboron compounds with B_2cat_2 without transition metal catalysts. This provides a new transformation pattern for the production of organoboron compounds. But the author already reported a reaction of B_2cat_2 , which gives similar products from aldehydes (ref. 25, *JACS* **2020**, 13011). Thus the present reaction is the carboxylic acid version of the previous *JACS* paper. Besides, this reaction requires a large excess of HMPA, which is a strong carcinogen, as a solvent and relatively high temperature (100 degrees C). The author mentions the advantage of metal-free conditions, but the reaction condition is a toxic and not environmentally benign process. From these points, it is difficult for me to see this paper has enough novelty and usefulness suitable for *Nat. Comm.*.

Response: Thank you for your comment on our work, which mainly focused on 1) similarity to our previous work; 2) high reaction temperature; 3) toxicity of HMPA solvent. Below, we would like to reply to these concerns point by point.

1) Similarity to our previous work

Our idea of using carboxylic acids as unconventional electrophiles in reductive borylation reaction was inspired by our prior efforts on deoxygenative borylation of aldehydes/ketones, where we suffered some intrinsic disadvantages due to the limited availability and poor stability of the latter carbonyl compounds. Therefore, we thought that using carboxylic acids, which benefit from being readily available, cost-effective, shelf-stable and structurally diverse, could advance our previous protocol for boronate synthesis.

During the exploration of this work, we noticed some different and interesting features between our current and previous work. Therefore, we made efforts and supplemented some comparison experiments.

a) Alkoxylated benzoic acids and heteroaromatic acids were generally ineffective in the previous protocol; however, under the new conditions, the yields of corresponding benzylboronates were significantly boosted, ranging from 2 to 10 times. We believed that this would be an important addition to the scope of deoxygenative borylation.

b) We were unable to convert the sterically hindered aldehydes/ketones such as those bearing α -quaternary carbon centres to the desired product. However, we found that the carboxylic acids and their sodium salts, which might undergo different elementary steps in the boronate-forming mechanism, could exhibit different reactivities. Taking **43b'** and **44b'** as examples, the corresponding aldehydes showed little to no yields, while both acids and sodium salts could give the 1,1-diboronate products.

c) The synthetic advantages of our new methods were demonstrated in preparing some small-molecule multi-boronates using some readily available light acid, for instance, acetic acid and formic acid. However, for the same synthetic purposes, conventional approaches often necessitated some gaseous substrates or some energetic compounds that were inconvenient to handle (1,1,2-trisborylethane: possibly from $H-C\equiv C-H$, *J. Am. Chem. Soc.* **136**, 16140-16143 (2014).; *Green Chem.* **19**, 3997-4001 (2017).; 1,1-bisborylmethane: from CH_4 , *Angew. Chem., Int. Ed.* **58**, 10671-10676 (2019).; from $C\equiv O$, *Angew. Chem., Int. Ed.* **55**, 4707-4710 (2016).; from CH_2N_2 , *Organometallics* **20**, 3962-3965 (2001).).

In particular, the tris[(pinacolato)boryl]ethane (**35b**), which has not been documented before, represented an important synthetic challenge and could potentially serve as a functionalised

ethylating reagent. Although it could be accessed from acetaldehyde (**35c**) using our previous protocol, such a transformation was inefficient due in part to the high volatility of low-weight aldehyde and some off-target reactivities, such as oligomerisation and condensation of acetaldehyde.

For the synthesis of bisboryl methane (**41b'**), our attempts to exploit formaldehyde, either in aqueous solution or paraformaldehyde form, were unsuccessful as our aldehyde conditions could not tolerate excessive water and polymeric substrates. Fortunately, simple formic acid was proved viable starting material under the new conditions.

d) Taking advantage of our acid method allowed us to synthesise several valuable isotope-labelled boronate building blocks using readily accessible and low-cost acid starting materials. These deuterated and ¹³C-labelled boronates could be diversified into various isotopic products that were useful in different contexts. However, simple and convenient routes to prepare the same boronate products still remained obscure in the literature. Arguably, these building blocks could not be easily obtained based on the known borylation protocols.

In principle, the same products could be obtained from the corresponding aldehyde, albeit at much higher costs (prices provided by Sigma Aldrich, and the prices for octadecanal-1-¹³C, hydrocinnamic acid-2-¹³C and octadecanal-d₃₅ were not found, therefore, not shown). It is worth mentioning that these aldehydes were mostly derived through multi-step synthesis involving the redox adjustment of the corresponding carboxylic acids.

From the above points, we wished to convey that the upgrade from aldehydes/ketones to carboxylic acids for boronate preparation is not a simple extension but leads to a more practical and enabling approach.

In addition, reductive functionalisation of carboxylic acids or their derivative in a controllable manner was difficult to achieve. Such difficulty could partially be attributed to the formation of unstable aldehyde intermediates, which would be soon reduced into alcohol or methylene as side products (*J. Am. Chem. Soc.* **142**, 8109-8115 (2020)). To relieve these unreactive end product (e.g., alcohols and methylenes for boronate synthesis, extra steps and reagents were required.

2) High reaction temperature

Direct deoxygenative of carboxylic acids to functionalised alkane ($R-CO_2H$ to $R-CH_2FG$) often demanded high thermal energy input because the carboxylic group was generally

thermodynamically stable and kinetically inert. Depending on the catalysed or non-catalysed systems, the reaction temperature generally ranged from 80 °C to 220 °C, and the examples below this range remained rare. For representative samples, please see below.

A | C-N bond formation: *Angew. Chem., Int. Ed.* **57**, 11673-11677 (2018).

B | C-O bond formation: *Angew. Chem., Int. Ed.* **54**, 4739-4749 (2018).

C | C-C bond formation: *Chem. Sci.* **8**, 6439-6450 (2017).

We attempted to decrease the reaction temperature, which unfortunately led to lower conversion and lower yields in both cases of free acids and their sodium salts.

3) Toxicity of HMPA solvent

In light of the detrimental effect of HMPA, we made several attempts to avoid its usage, including reducing its loading and testing other solvents.

A | Solvent screening for ArCO₂Na deoxy-borylation

solvent other than HMPA	yield (1b)
Other amide: DMA, NMP, DMPU, DMEU	10-38%
Alcohol: MeOH, i -PrOH, t -BuOH	Not detected
Ether: Et ₂ O, THF, dioxane, DME, diglyme	Not detected
Hydrocarbon: hexane, cyclohexane, benzene, toluene, Ph-CF ₃	Not detected
Chlorinated alkane: DCM, CHCl ₃ , DCE	Not detected
Others: DMSO, CH ₃ CN, EtOAc, acetone	Not detected

B | Solvent screening for ArCO₂H deoxy-borylation

solvent other than HMPA	yield (1b)
Other amide: DMA, NMP, DMPU	45-54%
Alcohol: MeOH	Not detected
Ether: THF	Not detected
Hydrocarbon: hexane, toluene, Ph-CF ₃	Not detected
Chlorinated alkane: DCM, CHCl ₃	Not detected
Others: DMSO, CH ₃ NO ₂ , CH ₃ CN, EtOAc, acetone, DMC	Not detected

C | ArCO₂H deoxy-borylation: HMPA vs DMA

solvent	concentration	equivalence	yield (1b)
HMPA	0.33 M	17.0	61%
HMPA	0.17 M	8.5	36%
DMA	0.33 M	32.0	54%
DMA	0.17 M	16.0	25%
DEA	0.33 M	24.0	17%

More than 25 common organic solvents in different concentrations and their combination in various ratios were examined in the deoxygenative borylation of 4-biphenylcarboxylic acid and its sodium salt. Some representative results were shown in the tables above. From these data, we could see some trends:

1) This diboron-mediated deoxygenation reaction exhibited a very strong and unique dependence on the nature of the solvent. Among all the solvents and their mixtures tested, the desired product was observed in reasonable yield only if the amide-typed solvent was present. This solvent effect could be rationalised by that a) exposing B₂cat₂ to a coordinating environment was indispensable for its activation and subsequent reactivity, and also proper solubility; b) some solvents were intrinsically incompatible with B₂cat₂, which possesses moderate Lewis acidity and

reducing capability. For instance, EtOAc and CH₃CN were prone to hydrolysis in the presence of B₂cat₂, and the oxidising DMSO were incompatible with our optimal conditions since DMS was observed.

2) Decreasing the amide solvent loading resulted in a lower yield of the benzylboronate product, presumably due to the less efficient formation of the amide-diboron adduct, which was proposed to be the key intermediate in this transformation.

Although most of our effort in minimising the involvement of HMPA or identifying a benign substituent of it was unsuccessful, DMA, which was commonly used as a chaperone with B₂X₄ in reductive borylation chemistry (*Science* **357**, 283-286 (2017); *Angew. Chem., Int. Ed.* **57**, 15227-15231 (2018). *Chem. Sci.* **10**, 161-166 (2019); *Org. Lett.* **22**, 234-238 (2020).) and was often seen in other types of reductive transformations, could still give a reasonable yield of **1b** (54%).

A quick spot check of some representative substrates showed that reactions in DMA could lead to comparable productivities to those in HMPA. Other examples involving complex carboxylic acid and some isotope-labelled substrates also showed promising results when DMA was employed as solvent.

During the substrate scope exploration with DMA, 1,1,2-trisborylated ethane **35b** was occasionally observed in small quantity, whose identity was confirmed by both NMR and GC-MS. The presence of such a side product indicated the background reaction between B₂cat₂ and DMA solvent, which might account for the generally lower yields of reactions conducted in DMA. Armed with this information, a more robust amide solvent, *N,N'*-diethylacetamide (DEA), was evaluated; however, it yielded less

desired product, possibly because it is too hindered to complex with B_2cat_2 (see last entry of the left Table C above).

Taken together, if the harmful effect of HMPA was a major concern when using our acid deoxygenative borylation protocol, especially for large-scale synthesis, we showed that DMA served as a more user-friendly yet analogously efficient surrogate for the HMPA in our deoxygenative borylation chemistry. Besides, exploring a more robust coordinating solvent, which was more resistant toward hydrolysis and reduction, would be a promising way to further improve the current protocol.

1. The reaction mechanism is still not clear. If the proposed reaction mechanism supposed by the authors can be employed (in Scheme 5A), the most important key step is **B1** or **B2** intermediate to acyl boron compound. This can be investigated by DFT calculations.

Response: Thank you for your comment on our work. Accordingly, density functional theory (DFT) calculations were carried out in order to gain insights into the mechanism of the acylboronate formation from intermediates **B1** and **B2**. Considering the cost of time, our computational effort mainly focused on this step.

1. Computational methods

Density functional theory (DFT) calculations were carried out in order to gain insights into the mechanism of the acylboronate formation from intermediates **B1** and **B2**. All calculations were performed using the Gaussian software package (version 16, revision B.01). The dispersion-corrected B3LYP approximation was used as the exchange-correlation functional. The 6-31++G(d,p) basis set was employed to represent spin-polarised molecular orbitals. DMA solvent was represented implicitly using the polarisable continuum model in the integral equation formalism. Structures of stable intermediates and transition states were fully optimised. Thermodynamic functions were calculated using the harmonic approximation for the standard-state temperature of 298.15 K and concentration of $1 \text{ mol} \times L^{-1}$. The stable intermediate structures and transition states were shown together with their standard-state Gibbs free energies.

2. Results of DFT modelling

2.1. Formation of the acylboronate intermediate from free carboxylic acid

Eq. 1

Eq. 2

Acetic acid (HOAc) was used as a simple and representative acid to model the formation of acylboronate from free carboxylic acids. The proposed mechanism of the formation of acylboronate intermediate is shown in the equation above. The Gibbs energy profile of the reaction is shown below.

2.2. Formation of the acylboronate intermediate from sodium carboxylate

Acetate anion (AcO^-) was used as a simple and representative carboxylate to model the formation of acylboronate from sodium carboxylate. The proposed mechanism of the formation of acylboronate intermediate is shown in the equation above. The Gibbs energy profile of the reaction is shown below.

3. Discussion of the DFT results

The calculations showed that all elementary steps are thermodynamically favourable in the cases of both the free acid and carboxylate anion. This indicated that the formation of the boron-oxygen bond is an effective driving force of the acylboronate formation. The height of the kinetic barriers

showed that the transformations are plausible. However, the activation energies appeared to be overestimated. Unlike stable molecules, transition state geometries and barrier heights are difficult to reproduce accurately using the exchange-correlation functionals, B3LYP in this case, that are fitted to experimental data collected for stable systems. Other possible sources of error are the Harmonic treatment of low-frequency vibrations and implicit treatment of solvent molecules. It can be speculated that polar HMPA molecules with negatively charged oxygen atom can form hydrogen or donor-acceptor bonds with the reacting species, which can be stronger in the transition state than reactants, thus lowering the activation energy.

Future more extensive simulations employing the quantum mechanics/molecular mechanics (QM/MM) method to describe solvent molecules explicitly and accelerated molecular dynamics to estimate the free energies beyond harmonic approximation can provide a more accurate description of the reaction and elucidate the nature of the observed solvent effect.

Additional experiments such as in-situ ^{13}C NMR monitoring was also performed to probe the presence of acylboronate intermediate. However, due to 1) limited availability of deuterated solvents, such as HMPA- d_{18} or DMA- d_9 , which led to poor locking and shimming; 2) intrinsically weak ^{13}C NMR signals of carbonyl carbons; 3) quadrupolar relaxation induced by boron atom, which collectively led to poor NMR response in our case, we failed to observe the formation of acylboronate even by using benzoic acid- α - ^{13}C . Interestingly, the corresponding aldehyde carbonyl signal (benzaldehyde- α - ^{13}C in this case) was also absent during the in-situ monitoring, which perhaps indicated that aldehyde was not the active intermediate in this transformation.

2. I don't feel several application studies in Scheme 3 were not variable and appealing. For Scheme 2B, not many researchers want to convert benzoic acid to 1-phenylacetic acid by the present procedure ($\text{B}_2\text{cat}_2/\text{HMPA}$; Cu cat/ CO_2). Similarly, preparations of **2b** and **38b'-H₂** are little to be gained for the effort. For making these processes more appealing, reactions including more complicated starting materials with functional groups to hard-to-obtain-products should be shown.

Response: Thank you for your comment on our work.

For the application shown in Scheme 3A of the manuscript, we aimed to highlight the difference between decarboxylative borylation ($-\text{CO}_2$, well explored) and our deoxygenative borylation ($-\text{[O]}$, underdeveloped) and apply it in combination with other known methods. As a proof-of-concept, some simple carboxylic acids ($\text{Ph-CO}_2\text{H}$ and $\text{Ph-CH}_2\text{-CO}_2\text{H}$) and boronic acid esters ($\text{Ph-CH}_2\text{-Bpin}$ and $\text{Ph-CH}_2\text{CH}_2\text{-Bpin}$) were chosen to demonstrate this one-carbon addition/reduction approach.

On the one hand, we agreed that these examples were not appealing enough for synthetic chemists. Nevertheless, when proceeding to more sensitive functional groups and complicated starting materials to illustrate the practicality of these methods, we were restricted by the intrinsic limitation of our borylation method and the narrow scope of Cu-catalysed Bpin-to-CO₂H transformation. More efforts were needed to mature this homologation strategy.

On the other hand, we performed further application-focused experiments to show the applicability of our acid deoxygenative borylation method.

1) The isotope labelled boronates

Taking advantage of our acid method allowed us to synthesise several valuable isotope-labelled boronate building blocks using readily accessible and low-cost acid starting materials. These deuterated and ¹³C-labelled boronates could be diversified into various isotopic products that were useful in different contexts. Arguably, these building blocks could not be easily obtained based on the known borylation protocols, and simple and convenient routes to prepare the same boronate products still remained obscure in the literature.

In principle, the same products could be obtained from the corresponding aldehyde, albeit at much higher costs (prices provided by Sigma Aldrich, and the prices for octadecanal-1-¹³C, hydrocinnamic acid-2-¹³C and octadecanal-*d*₃₅ were not found, therefore, not shown). It is worth mentioning that these aldehydes were mostly derived through multi-step synthesis involving the redox adjustment of the corresponding carboxylic acids.

2) Using contaminated aldehyde/carboxylic acid as starting material

In our original manuscript, we demonstrated that our deoxy-borylation method could convert mixed aldehyde and acid into the same boronate product, and their ratio in the mixture was inconsequential. These results led to our later discovery that the crude of alcohol and methylarene oxidation reaction, which consisted of partial oxidant products in various ratios, could be directly used in subsequent borylation reaction, formulating the one-pot, two-step alkylboronate synthesis.

The facile autooxidation of aldehyde mandated careful storage and purification prior to usage. To exhibit the practical value of our method, an authentic sample of acid-contaminated *p*-tolualdehyde (in 1H NMR, $RCHO:RCO_2H = 4:1$) was directly subjected to the deoxygenative borylation, and the corresponding alkylboronate (**3b**) could be obtained exclusively in good yield, circumventing the time-consuming distillation.

A | Deoxy-borylation of mixed aldehyde and acid

B | Tandem alcohol oxidation & deoxy-borylation

C | Tandem methylene oxidation & deoxy-borylation

D | Deoxy-borylation of authentic acid-contaminated aldehyde

3) Structurally complex starting materials

In probing the robustness of our conditions, several carboxylic acids derived from the natural product, drug molecule and others with complex scaffolds were subject to our deoxygenative borylation. Encouragingly, all of them could be borylated smoothly, giving the monoboronates and tris(boronate) with diverse molecular complexity.

In particular, benzylboronate **45g** with a bulky and coordinating phosphino substituent could be useful in designing some phosphine ligands and catalysts; however, its synthesis could cause problems in traditional metal-involved approaches, emphasising the advantages of our metal-free strategy.

4) Synthesis of bioactive or other synthetically useful compounds

The alkylboronates directly derived from carboxylic acids under our optimal conditions were found useful in preparing some complex molecules that were interesting in the pharmaceutical and material industry. In contrast, to access the same boronate building block, existing methods often involved multiple synthetic steps when using carboxylic acids as raw materials.

In combination with some well-established downstream transformations, our current conditions could formulate a simple and modular approach to build up molecular complexity efficiently. For examples, using the trifluoroborates derived from carboxylic acids, a deoxygenative borylation/Suzuki-Miyaura coupling process could be iterated to access a multi-benzylated aldehyde, which was proven useful intermediate for paracyclophane synthesis. In other cases, the modularity of benzylboronate synthesis using our acid deoxygenative borylation method allowed us to easily prepare several precursors or derivatives of pharmaceutical candidates, e.g., OMT inhibitor and drug for arenavirus treatment.

A | Synthesis of cycloparaphane

B | Synthesis of drug molecules

Taken together, we hope that the reviewer's concerns on 1) conceptual similarity to our prior efforts; 2) unbenign reaction conditions; 3) insufficient mechanistic support (by computation); 4) limited application examples; were addressed, and we wished the reviewer could re-evaluate our work and consider its publication on *Nature Communication*.

Reviewers' Comments:

Reviewer #1:

Remarks to the Author:

The authors took the revision process very seriously and improved the quality and scholarly presentation of the manuscript substantially. Publication in Nature Communication is highly recommended.

Reviewer #2:

Remarks to the Author:

I think the manuscript has been well revised. The raised concerns have been addressed via additional experiments and discussions. Therefore, it is suitable for publication.

By the way, I think the following report about the subject "alkyl decarboxylative borylation" should be included, at least in the references.

- Org. Lett. 2017, 2770–2773.

Reviewer #3:

Remarks to the Author:

The point discussed in the previous comments is that the current reaction of deoxygenative borylation of carboxylic acids is essentially the same as the previous reaction of aldehydes that they reported in JACS. An additional part of this paper to the previous JACS paper is the reduction of carboxylic acids to acylboron intermediates. After the carboxylic acid becomes an acylboron intermediate, the reaction is the same as described in the previous JACS paper. Although it is well understood that the availability and handling stability of carboxylic acids are higher than those of aldehydes, the advances made in this paper are merely technical improvements and not new advances worthy of publication in top-level journals. Many additional experiments and discussion are added in this revision, but this paper still has essential issues.

Reviewer 1

The authors took the revision process very seriously and improved the quality and scholarly presentation of the manuscript substantially. Publication in Nature Communication is highly recommended.

Response: Thank you for your comment on our work.

Reviewer 2

I think the manuscript has been well revised. The raised concerns have been addressed via additional experiments and discussions. Therefore, it is suitable for publication.

Response: Thank you for your comment on our work.

By the way, I think the following report about the subject "alkyl decarboxylative borylation" should be included, at least in the references (Org. Lett. 2017, 2770–2773.)

Response: Thank you for your suggestion, and we have added this paper in our reference list.

Reviewer 3

The point discussed in the previous comments is that the current reaction of deoxygenative borylation of carboxylic acids is essentially the same as the previous reaction of aldehydes that they reported in JACS. An additional part of this paper to the previous JACS paper is the reduction of carboxylic acids to acylboron intermediates. After the carboxylic acid becomes an acylboron intermediate, the reaction is the same as described in the previous JACS paper. Although it is well understood that the availability and handling stability of carboxylic acids are higher than those of aldehydes, the advances made in this paper are merely technical improvements and not new advances worthy of publication in top-level journals. Many additional experiments and discussion are added in this revision, but this paper still has essential issues.

Response: Thank you for your comment on our work. We agreed that the current work shares some similarities to our previous one, from which we got inspiration. In our last response, we included additional experiments and discussion, intending to show that switching from aldehyde to carboxylic acid is beyond "technical improvement".

1) Apart from the increased availability and handling convenience, using carboxylic acid as starting materials led to broader scope with the practical application since the corresponding aldehyde is inaccessible or incompatible with deoxygenative borylation chemistry. Indeed, our aldehyde-based method often shows complementary reactivity to the current one.

2) Conceptually, the deoxygenative functionalization of aldehyde and carboxylic acid represent two different synthetic challenges. It is more than cleaving one more C-O bond but mandating a robust reducing system with balanced reactivity since the acid substrate, and side products (e.g., alcohol and methylene) are "inert"; however, the acyl intermediate is highly reactive.

3) We understand that our method still suffers some limitations; however, we believe that they could encourage our future research endeavours.